# *Syzygium gratum* Extract Alleviates Vascular Alterations in Hypertensive Rats

**DOI:** 10.3390/medicina56100509

**Published:** 2020-09-30

**Authors:** Poungrat Pakdeechote, Sariya Meephat, Chadamas Sakonsinsiri, Jutarop Phetcharaburanin, Sarawoot Bunbupha, Putcharawipa Maneesai

**Affiliations:** 1Department of Physiology, Faculty of Medicine, Khon Kaen University, Khon Kaen 40002, Thailand; ppoung@kku.ac.th (P.P.); sariya_m@kkumail.com (S.M.); 2Research Institute for Human High Performance and Health Promotion, Khon Kaen University, Khon Kaen 40002, Thailand; 3Department of Biochemistry, Faculty of Medicine, Khon Kaen University, Khon Kaen 40002, Thailand; schadamas@kku.ac.th (C.S.); jutarop@kku.ac.th (J.P.); 4Faculty of Medicine, Mahasarakham University, Maha Sarakham 44000, Thailand; sarawoot.b@msu.ac.th

**Keywords:** *Syzygium gratum*, hypertensive rats, L-NAME, vascular dysfunction and hypertrophy, renin angiotensin system

## Abstract

*Background and Objectives*: *Syzygium gratum* (SG) is a local vegetable and widely consumed in Thailand. Previously, a strong antioxidative effect of SG extract has been reported. The effects of SG extract on hypertension have remained unknown. The effect of SG aqueous extract on blood pressure and vascular changes were examined in L-NAME-induced hypertensive rats (LHR), and its potential active constituents were also explored. *Materials and Methods:* Male Sprague Dawley rats were allocated to control, L-NAME (40 mg/kg/day), L-NAME + SG (100, 300, 500 mg/kg/day), or captopril (5 mg/kg/day) groups. The components of SG extract were analyzed. *Results:* The analysis of aqueous SG extract was carried out using HPLC-Mass spectroscopy, and phenolic compounds could be identified as predominant components which might be responsible for its antihypertensive effects observed in the LHR model (*p* < 0.05). Additionally, SG extract also improved vascular responses to acetylcholine and decreased vascular remodeling in LHR (*p* < 0.05). Enhancements of eNOS expression and plasma nitric oxide metabolite levels, and attenuation of angiotensin converting enzyme (ACE) activity and plasma angiotensin II levels were observed in the LHR group treated with SG. Moreover, SG exhibited strong antioxidant activities by reducing vascular superoxide generation and systemic malondialdehyde in LHRs. Captopril suppressed high blood pressure and alleviated vascular changes and ACE activity in LHRs, similar to those of the SG extract (*p* < 0.05). *Conclusion:* Our results suggest that the SG extract exhibited antihypertensive effects, which is relevant to alleviation of vascular dysfunction and vascular remodeling of LHRs. These effects might be mediated by phenolic compounds to inhibit ACE activity and scavenge reactive oxygen species in LHR.

## 1. Introduction

In general, nitric oxide (NO) is synthesized from vascular endothelium to mediate vascular smooth muscle relaxation and vascular protective effects [1]. NO deficiency induced by N-nitro-L-arginine-methyl ester (L-NAME), an L-arginine analogue, can enhance systemic vasoconstriction and arterial hypertension in rats [2]. Furthermore, other cardiovascular complications, such as cardiac dysfunction, heart failure, vascular dysfunction, and hypertrophy in long-term treatment of L-NAME in animal models have been noted [3,4]. Impairment of endothelium-dependent vasodilation in rats induced by L-NAME linked with reductions of eNOS protein expression and plasma nitrate/nitrite concentrations have been shown [5]. Moreover, aortic hypertrophy, including increases in vascular wall thickness, cross-sectional area, and wall-to-lumen ratio occurred after chronic L-NAME administration was shown [6,7]. Renin angiotensin system (RAS) activation characterized by increases in angiotensin converting enzyme activity and systemic angiotensin II (Ang II) concentration has also been observed in hypertensive rats with NO depletion [3,8]. This RAS activation is a consequence of renal vasoconstriction caused by L-NAME [9]. It has been known that Ang II is a potent vasoconstrictor and mediates cardiovascular hypertrophy [10]. Furthermore, Ang II might be a major contributor of oxidative stress in L-NAME hypertension [11].

Previous studies demonstrated that oxidative stress plays an important role in the pathology of hypertension [12,13,14]. In L-NAME hypertensive rats, oxidative stress has been reported to increase lipid and protein peroxidation, plasma malondialdehyde (MDA) levels, and protein carbonyl content [15]. Furthermore, vascular superoxide production is high in L-NAME-treated rats [16]. It is well-established that increases in reactive oxygen species can quench systemic NO bioavailability [17] and subsequently impair vascular function. Therefore, various substances with antioxidant properties could be considered to cause the enhancement of NO bioavailability and the suppression of hypertension.

*Syzygium gratum* (SG) (S. gratum, synonym: *Eugenia grata* Wight) of theMyrtaceae family has been known as a local vegetable in South-East Asia. The nutraceutical merit of its leaves has been reported to contain a high content of protein, beta-carotene, and iron. The biological effects of SG extract has been demonstrated in both in vivo and in vitro studies. Kukongviriyapan and coworkers found that SG extract had high free radical scavenging and antioxidant activities and alleviated phenylhydrazine-induced vascular alterations in rats [18]. The SG aqueous extract had relatively high antioxidant activity assayed by ferric-reducing antioxidant power and total phenolic assay and increased the cytoprotective enzyme, heme oxygenase (HO-1) activity, as well as the expression of HO-1 mRNA in C57BL/6J mice [19]. SG extract was speculated to have potential anti-cancer activity, due to its cytotoxic effect on cancer cells in vitro [20]. A recent study showed that phenolic compounds in SG methanolic extract might mediate cytotoxic activity in MCF-7 breast adenocarcinoma and MDA-MB-231 breast cancer cell lines [21]. This current study was designed to investigate the effect of SG aqueous extract on blood pressure, vascular dysfunction, and hypertrophy in hypertensive rats induced by a NOS inhibitor.

## 2. Materials and Methods

### 2.1. Animals and Experimental Protocol

The present study was approved by the Animal Ethics Committee of Khon Kaen University, Khon Kaen, Thailand (ACUC-KKU-29/60), and followed the guidelines for the care and use of experimental animals. Experiments were performed with male Sprague-Dawley rats weighing 200–250 g (Nomura Siam International Co., Ltd., Bangkok, Thailand), housed under controlled environmental conditions (25 ± 2 °C, 12 h light/dark cycle) with free access to food and water. Before starting the experiment, all rats were acclimatized for seven days and then were randomly allocated into six experimental groups (*n* = 8/each) including control, L-NAME, L-NAME+SG extract (100, 300, or 500 mg/kg/day), and L-NAME + 5 mg/kg/day captopril. For five weeks of the study, the control rats were given tap water, while the hypertensive rats were given L-NAME (Sigma-Aldrich, St. Louis, MO, USA, 40 mg/kg/day) in their drinking water to induce hypertension. Various doses of SG extract, vehicle, or captopril were orally administrated every day for the last two weeks of the study period.

### 2.2. Preparation of SG Extract and Chemicals Analysis of the Extract

The fresh SG leaves were cultivated in local agricultural fields of the Khon Kaen province, Thailand, and the leaves were identified as SG by the same method as previously described by Kukongviriyapan et al., in 2007 [18]. After that, the leaves were weighed, chopped, and boiled in distilled water for 30 min. Thereafter, the extract was dried using a spray-drying machine, which yielded 16.7% SG [19]. Biochemical components of SG extract were analyzed using reversed phase ultra-high-performance liquid chromatography coupled with quadruple time-of-flight mass spectrometry (RP-UHPLC-QTOF-MS).

### 2.3. Measurement of Systolic Blood Pressure

To monitor changes in blood pressure induced by L-NAME, systolic blood pressures (SBP) and heart rates (HR) were determined in all rats using tail-cuff methods (IITC/Life Science Instrument model 229 and model 179 amplifier (Woodland Hills, CA, USA)) throughout the five weeks of the experimental period.

### 2.4. Vascular Function Study

At the end of the experiment, the rats were anesthetized and killed by exsanguination. The thoracic aortas were removed and placed in Kreb’s solution. After cleaning off connective tissue, the aortas were cut into rings for tension measurement. To examine the vascular response to various vasoactive agents, the rings were pre-contracted with phenylephrine (10 µM) before acetylcholine (ACh, 0.01 µM–3 µM) or sodium nitroprusside (SNP, 0.01 µM–3 µM) were cumulatively applied with various concentrations in the baths. The results are expressed as percent relaxation from the phenylephrine-induced contraction.

### 2.5. Measurement of Oxidative Stress Markers and Plasma Nitrate/Nitrite

Superoxide (O_2_^•−^) generated in the carotid artery was assessed using the lucigenin-enhanced chemiluminescence method following the previously published method [16]. Plasma NOx levels were determined using an enzymatic conversion method reported by Verdon et al., 1995 [22] with some modification [16] to evaluate plasma NOx levels. Firstly, the plasma protein was filtrated using nanosep (Pall Life sciences, Portsmouth, UK) and the filtrate (80 µL) was collected and mixed with NADPH, G-6-P, G-6-PD and nitrate reductase, and then incubated at 30 °C for 30 min. The reaction mixture was mixed with a Griess solution (100 µL), and kept at room temperature for 15 min. Then, the absorbance at 540 nm was measured using a microplate reader (Tecan GmbH., Groding Australia). The serial dilution of NaNO_2_ stock solution was carried out to prepare standard solutions reported previously [16].

### 2.6. Morphometric Measurement

The thoracic aortas at the third to sixth vertebral levels were separated and fixed with 4% paraformaldehyde. The tissues were processed and embedded in paraffin and cut before staining with hematoxylin and eosin. Images were viewed under a DS-2Mv light microscope (Nikon, Tokyo, Japan). Morphometric evaluations and vascular smooth muscle cell numbers (VSMC) were analyzed and counted under Image J morphometric software (National Institutes of Health, Bethesda, MD, USA).

### 2.7. Measurement of Angiotensin Converting Enzyme (ACE) Activity and Serum Angiotensin II (Ang II) Concentration

Serum ACE activity was determined using a fluorescence assay, as previously described [23], with minor modifications. The reaction mixtures containing the serum mixed with hippuryl-L-histidyl-L-leucine (HHL) in assay buffer were incubated at 37 °C for 30 min. After that, NaOH was added to stop the reaction and the product of the reaction was fluorogenically labeled with o-phthaldialdehyde (OPA). The fluorescence was read at 355 nm excitation; 535 nm emission using a fluorescent plate reader. The activity of ACE was reported as mU/mL. In addition, an Ang II Enzyme immunoassay (EIA) kit (St. Louis, MO, USA) was used to determine the level of serum Ang II.

### 2.8. eNOS Protein Expression Assay

Thoracic aortas were collected to determine the expression of eNOS protein in the tissues following a previous method [16]. The expression of β-actin was also determined as an internal standard.

### 2.9. Statistical Analysis

Results were expressed as the mean ± the standard error (SEM) and the statistical analysis was carried out using GraphPad prism 8.3 software (San Diego, CA, USA). One-way analysis of variance (ANOVA) followed by a post-hoc Turkey’s test was performed to test the differences between groups. *p* < 0.05 was considered to be statistically significant.

## 3. Results

### 3.1. Biochemical Components of SG Extract

Components of aqueous SG extract were analyzed using reverse phase ultra-high-performance liquid chromatography coupled with quadruple time-of-flight mass spectrometry (RP-UHPLC-QTOF-MS). A total of 2048 signals were detected in the positive electrospray ionization mode and 374 signals in the negative electrospray ionization mode. Background signals were removed by use of the coefficient of variation of less than 30%, and metabolite identification was then performed. The main compounds in the aqueous extract of SG are listed in Table 1.

### 3.2. SG Extract Alleviated Systolic Blood Pressure and Heart Rates in LHR

Adding L-NAME in drinking water for five weeks gradually caused the elevation of systolic blood pressure in LHRs compared to control rats (190.49 ± 4.2 vs. 117.42 ± 1.58 mmHg, *p* < 0.05). Administration of SG extract markedly reduced systolic blood pressure at doses of 100 (166.33 ± 3.68 mmHg), 300 (150.6 ± 0.76 mmHg), and 500 mg/kg/day (148.21 ± 1.11 mmHg) or captopril (144.83 ± 4.38 mmHg) (*p* < 0.05, Figure 1). In addition, LHR caused a rise from 425.19 ± 4.51 to 549.94 ± 12.38 beats/min (*p* < 0.05) at week five. SG extract at 100 (481.72 ± 24.76 beats/min), 300 (455.86 ± 1012 beats/min), and 500 mg/kg/day (471.73 ± 13.64 beats/min) and captopril treatment reduced HR in LHR compared to untreated rats at week five (474.75 ± 19.56 beats/min, *p* < 0.05).

### 3.3. SG Extract Improved Vascular Responses in Aortic Rings

The vasorelaxation response of aortic rings to ACh (0.01–3 mM) was significantly attenuated in LHRs compared to control rats (3µM, 11.26 ± 5.81 vs. 84.01 ± 3.22%) (*p* < 0.05). The response to ACh was improved with SG 100 (29.47% ± 5.15%), 300 (34.78 ± 3.78%) and 500 mg/kg/day (40.16 ± 5.70%) or captopril supplementation (33.53 ± 5.53%) compared to LHRs *p* < 0.05) (Figure 2A). The vasorelaxation responses to SNP were almost similar in all groups (Figure 2B).

### 3.4. SG Extract Enhanced Expression eNOS Protein and Plasma Nitrate/Nitrite Level

A significant decrease of eNOS protein expression was detected in aortas from LHRs compared to those of control rats (*p* < 0.05). Treated with SG extract at a dose of 300 mg/kg/day or captopril at a dose of 5 mg/kg/day markedly raised expression of the eNOS protein compared to LHRs (*p* < 0.05) (Figure 3A). In addition, a low level of plasma NOx was found in LHRs compared to the control rats (4.09 ± 0.63 vs. 11.10 ± 1.20 µM). This reduction was improved by SG extract and captopril supplementation (9.40 ± 1.50 and 9.61 ± 0.76 µM, Figure 3B).

### 3.5. SG Extract Suppressed Vascular Remodeling of Thoracic Aorta

Significant increases in the indices of vascular remodeling, such as the cross-sectional area (CSA), wall thickness, lumen diameter, wall thickness/lumen ratio, and VSMC were observed in thoracic aortas obtained from LHRs (Figure 4A–F, *p* < 0.05). Interestingly, SG extract or captopril treatment alleviated thickenings of aortic walls, CSA, and VSMC compared with LHRs (*p* < 0.05).

### 3.6. SG Extract Reduced Plasma Ang II Level and Serum ACE Activity

A high level of serum ACE activity was observed in LHRs compared to those of control rats (180.62 ± 15.45 vs. 90.12 ± 8.15 mU/mL, *p* < 0.01). Treatment with SG extract or captopril prevented the L-NAME-induced increase in serum ACE activity (82.96 ± 17.43 mU/mL, 81.42 ± 15.46 mU/mL, Figure 5A). Moreover, plasma Ang II was markedly increased in LHRs compared to the control rats (14.89 ± 2.56 vs. 5.99 ± 1.20 pg/mL, *p* < 0.01). SG extract (300 mg/kg/day, 8.59 ± 0.69 pg/mL) and captopril (7.54 ± 0.92 pg/mL) significantly decreased concentrations of plasma Ang II compared to LHRs (Figure 5B).

### 3.7. Effect of SG Extract on Markers of Oxidative Stress

Vascular O_2_^•−^ generation was high in LHRs compared to control rats (230.59 ± 10.78 vs. 40.25 ± 4.49 count/mg dry wt/min, *p* < 0.05). SG extract in BW/day (100, 61.29 ± 4.42, 300, 48.49 ± 4.35, 500 mg/kg/day, 59.85 ± 7 count/mg dry wt/min), and captopril at 5 mg/kg/day (54.79 ± 3.00 count/mg dry wt/min) significantly decreased vascular O_2_^•−^ generation compared to LHRs, *p* < 0.05) (Figure 6A). Subsequently, plasma MDA was raised in LHRs compared with the control rats (11.50 ± 0.76 vs. 3.18 ± 0.38 µM, *p* < 0.05). Interestingly, SG extract or captopril restored the L-NAME-induced high levels of plasma MDA compared to LHRs (6.33 ± 0.82, 3.59 ± 0.41, 3.96 ± 0.36 and 3.24 ± 0.32 µM, *p* < 0.05, Figure 6B).

## 4. Discussion

This study suggests that the major components of SG extract might be phenolic compounds. SG extract caused the reduction of blood pressure in a dose-dependent manner in the hypertensive rat model produced by chronic L-NAME administration. Endothelial dysfunction characterized by a decrease in vascular response to Ach and aortic hypertrophy were observed in aortic rings prepared from hypertensive rats. These vascular alterations were attenuated by SG extract in LHRs. SG extract also increased eNOS protein expression and plasma NOx in LHRs. RAS activation, including increases in serum ACE activity and plasma Ang II in LHRs, was suppressed by the SG extract. SG also exhibited an antioxidant property by reducing vascular superoxide generation and lipid peroxidation products, plasma MDA in LHRs. Captopril was used as a positive control, and it exerted antihypertensive effects and restored aortic dysfunction and hypertrophy relevant to reducing RAS activation and oxidative stress in LHRs.

Several biochemical compounds were identified in the SG extract, and most of them were phenolic compounds. Phenolic compounds have been known as secondary metabolites isolated from most food plants [24]. There are several subgroups of phenolic compounds characterized by their chemical structures, such as phenolic acids, flavonoids, tannins, coumarins, lignans, quinones, stilbens, and curcuminoids [25,26]. In this study, flavonoid bioactive substances listed in Table 1 were shown to be abundant in SG extract. Flavonoids, a large class of polyphenols, have contributed to health maintenance, since they exert antioxidant, inflammatory, anti-hypertensive, and anti-cancer activities [27,28]. Thus, the biological effects of SG extract on L-NAME-induced high blood pressure in rats will be discussed further.

LHR developed high blood pressure and increased HR that were associated with impairment of vascular responses to ACh and aortic hypertrophy. Blood pressure is influenced by two main factors—cardiac output, and vascular resistance. In LHR, reduction of NO generation, resulting in alterations of vascular endothelial function and an increase in peripheral vascular resistance have been established [29,30]. Sympathetic overactivity induced by chronic inhibition of NO synthesis by L-NAME has been suggested to raise heart rate in this animal model [31,32]. Aortic hypertrophy is caused by the vascular adaptive response to long-term hemodynamic changes, especially high blood pressure [33]. Furthermore, lacking NO has been proposed to promote vascular structural changes in hypertension [34]. Activation of RAS, such as high levels of serum Ang II, renin, ACE activity, and AT_1_R expression in LHR has been observed, and has contributed to hypertension and vascular remodeling in this animal model [35,36]. The current study findings showed that endothelial dysfunction and aortic hypertrophy were the consequence of decreased eNOS expression and plasma NOx, as well as increased serum ACE activity and ang II in LHR. Furthermore, enhancement of superoxide generation from eNOS uncoupling induced by L-NAME was revealed, and this ROS subsequently quenched NO to become ONOO^-^, a powerful oxidizing agent.

SG extract decreased systolic blood pressure and HR associated with improvement of vascular alterations in LHR. It restored expression of protein, eNOS, and NO metabolites in LHRs. Additionally, SG extract also suppressed the elevation of ACE activity and serum Ang II, as well as oxidative stress markers in LHRs. With the suppression of RAS activation, Ang II levels observed in this study might have been due to the ACE inhibitory activity of the SG extract. The ACE inhibitor property of SG extract is likely to be mediated by its main components, phenolic compounds, as mentioned above. There is a study to demonstrate the inhibitory effect of phenolic compounds on ACE activity suggesting that the carboxylate and hydroxyl groups can bind to the zinc ion at the active site of ACE and stop the enzyme reaction [37]. The bioactive molecules in the SG extract that were involved in the antioxidant effects might have been phenolic compounds, especially flavonoids, such as pinostrobin, pinocembrin, epigallocatechin-3-Gallate, neoliquiritin, and epigallocatechin. It has been known that flavonoids have the ability to reduce free radical formation and to scavenge free radicals [38]. The current results were accompanied by a previous study demonstrating that SG extract has potent antioxidant capacity using the FRAP and DPPH assays [19]. This antioxidant property of SG extract can prevent the reaction of O_2_^•−^ and NO and raise systemic NO levels to reduce vascular resistance and blood pressure. Captopril produced an antihypertensive effect via alleviation of vascular dysfunction and remodeling in LHRs. Additionally, captopril also reduced oxidative stress and RAS activation in LHRs. Therefore, it is possible that the SG extract had an antihypertensive effect, probably through the same mechanism as captopril.

## 5. Conclusions

SG extract mainly contained phenolic compounds. SG extract had anti-hypertensive and vascular effects that were allied with restorations of eNOS protein expression and plasma nitrate/nitrite, RAS activation, and oxidative status in LHR.

## Figures and Tables

**Figure 1 medicina-56-00509-f001:**
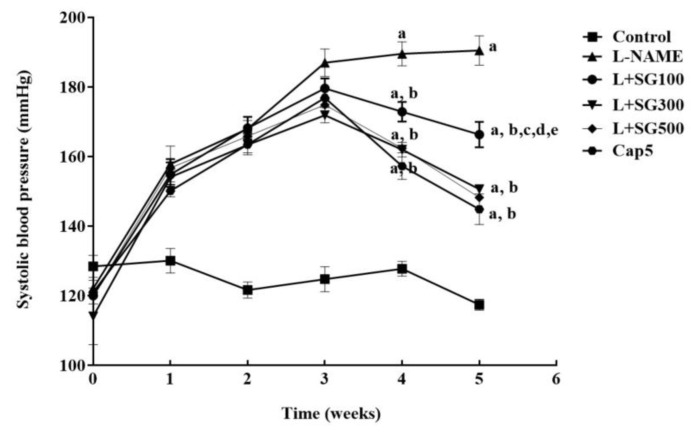
Changes of systolic blood pressure during five weeks of experiments in all groups of rats. Results are represented as mean ± SEM (*n* = 6–8). ^a^ < 0.05 vs. control group, ^b^ < 0.05 vs. L-NAME group, ^c^ < 0.05 vs. L + 100 group, ^d^ < 0.05 vs. L + 300 group, ^e^ < 0.05 vs. L + Cap5 group.

**Figure 2 medicina-56-00509-f002:**
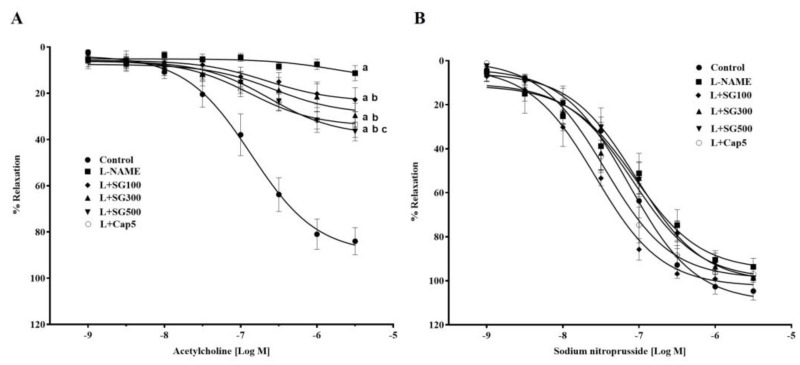
Effect of SG extract on vascular responses to exogenous acetylcholine (**A**), and sodium nitroprusside (**B**) in the thoracic aortas of all groups. Results are shown as mean ± SEM (n = 6–8/group), ^a^ < 0.05 vs. control, ^b^ < 0.05 vs. L-NAME, ^c^ < 0.05 vs. L + SG100.

**Figure 3 medicina-56-00509-f003:**
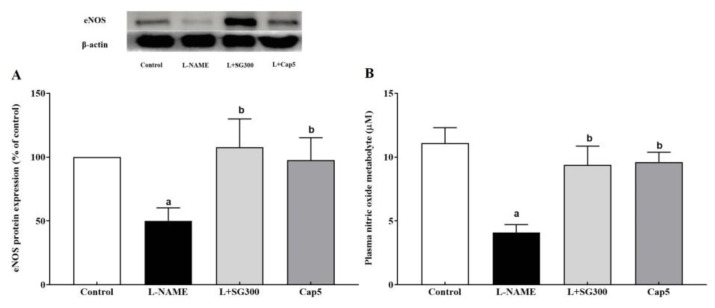
Effect of SG extract on plasma NOx (**A**) and eNOS protein expression (**B**) in the thoracic aorta. Data are expressed as mean ± SEM, (*n* = 6–8/group), ^a^ < 0.05 vs. control group, ^b^ < 0.05 vs. L-NAME group.

**Figure 4 medicina-56-00509-f004:**
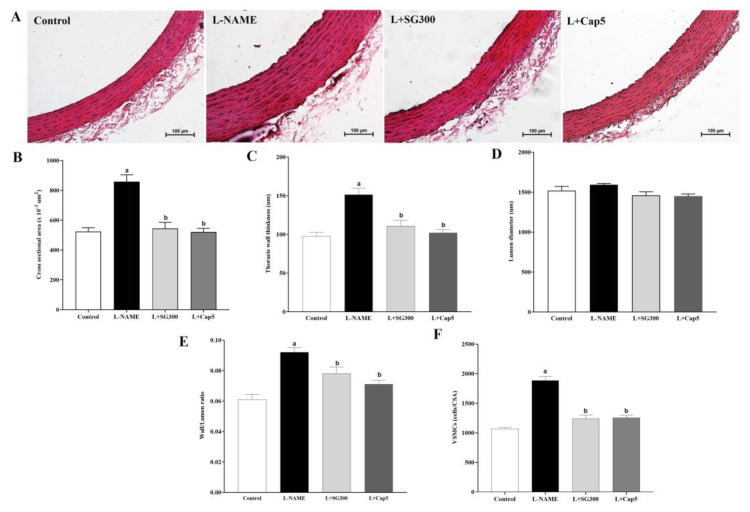
Morphological changes in aortic hypertrophy in all groups of rats. Illustrative images of thoracic aortas stained by hematoxylin and eosin, original magnification = X 20 (**A**) and values of cross-sectional areas (**B**), thoracic wall thicknesses (**C**), Luminal diameter (**D**), wall/lumen ratios (**E**) and vascular smooth muscle cell numbers (**F**) (*n* = 6). Data are mean ± SEM, *p* < 0.05, ^a^ < 0.05 vs. control group, ^b^ < 0.05 vs. L-NAME group.

**Figure 5 medicina-56-00509-f005:**
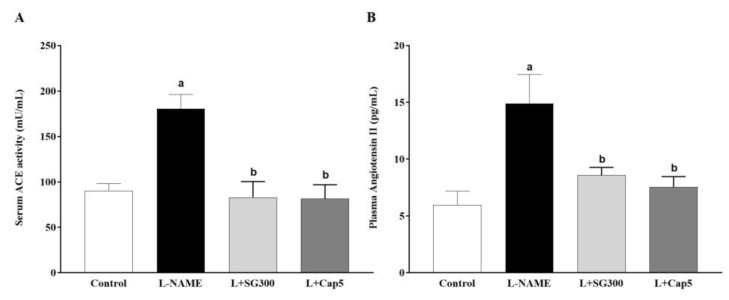
Serum angiotensin converting enzyme (ACE) activity (**A**) and plasma angiotensin II levels (**B**) in all rats (*n* = 6–8). Data are mean ± SEM, *p* < 0.05. ^a^ < 0.05 vs. control rats, ^b^ < 0.05 vs. L-NAME rats.

**Figure 6 medicina-56-00509-f006:**
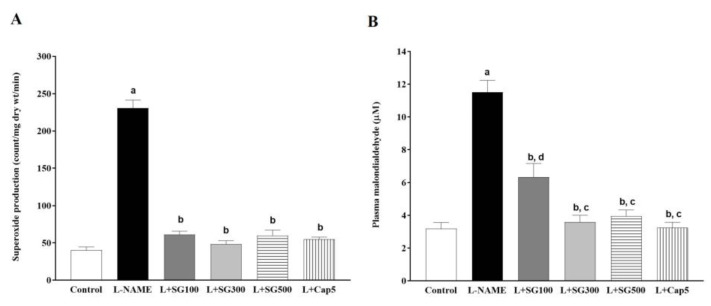
Effect of SG extract on vascular superoxide generation (**A**), and levels of plasma malondialdehyde (MDA) (**B**) in rats. Data are expressed as mean ± SEM, (*n* = 6–8/group), *p* < 0.05, ^a^ < 0.05 vs. control group, ^b^ < 0.05 vs. L-NAME group, ^c^ < 0.05 vs. L + SG100.

**Table 1 medicina-56-00509-t001:** Major components of aqueous *Syzygium gratum* (SG) extract analyzed using RP-UHPLC-QTOF-MS.

Retention Time	[M+H]^+^ (*m*/*z*)	Identified Compound	Formula	Class of Phytochemical
10.78	274.27	C16 Sphinganine	C_16_H_35_NO_2_	Phospholipid
12.52	271.10	Pinostrobin	C_16_H_14_O_4_	Flavonoids
11.7	257.08	Pinocembrin	C_15_H_12_O_4_	Flavanone
8.52	419.13	Neoliquiritin	C_21_H_22_O_9_	Flavonoids
6.65	459.09	Epigallocatechin-3-Gallate	C_22_H_18_O_11_	Flavonoids
8.51	257.08	Aloe emodin anthrone	C_15_H_12_O_4_	Anthraquinone
5.63	205.10	d-Tryptophan	C_11_H_12_N_2_O_2_	Amino acid
5.91	307.08	Epigallocatechin	C_15_H_14_O_7_	Flavonoids
7	481.10	Myricetin 3-glucoside	C_21_H_20_O_13_	Flavonoid-3-o-glycosides
1.35	268.10	Adenosine	C_10_H_13_N_5_O_4_	Purine nucleoside base
7.17	617.11	Quercetin 4-6-galloylglucoside	C_28_H_24_O_16_	Flavonol
7.59	319.04	Myricetin	C_15_H_10_O_8_	Flavonol
7.37	465.10	Myricitrin	C_21_H_20_O_12_	Flavonoid glycoside
10.18	297.24	Dimorphecolic acid	C_18_H_32_O_3_	Endogenous fatty acid
7.91	449.11	Quercitrin	C_21_H_20_O_11_	Flavonol
8.42	273.08	Naringenin	C_15_H_12_O_5_	Flavanone
6.64	291.09	Epicatechin	C_15_H_14_O_6_	Flavonoids
14.75	277.18	Buddledin A	C_17_H_24_O_3_	Terpenoids
13.32	239.13	2,2,4,4-Tetramethyl-6-1-oxopropyl-1,3,5-cyclohexanetrione	C_13_H_18_O_4_	Lipid
7.43	443.10	Catechin 7-O-gallate	C_22_H_18_O_10_	Flavonoid-7-o-glycosides
7.69	265.14	Abscisic Acid	C_15_H_20_O_4_	Isoprenoid plant hormone
14.88	473.36	Maslinic Acid	C_30_H_48_O_4_	Triterpenoids
10.36	295.23	12,13-Epoxy-9,15-octadecadienoic acid	C_18_H_30_O_3_	Long-chain fatty acids
8.1	229.09	cis-Resveratrol	C_14_H_12_O_3_	Polyphenol
14.37	478.29	Lysophosphatidylethanolamine (18:2)	C_23_H_44_NO_7_P	Lipid
5.82	655.19	Rhamnazin 3-sophoroside	C_29_H_34_O_17_	Flavonoid glycoside
5.19	627.15	6-Hydroxyluteolin 7-gentiobioside	C_27_H_30_O_17_	Flavonoid
11.09	281.17	Brefeldin A	C_16_H_24_O_4_	Lactone

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
