# Peer review of "Syzygium gratum Extract Alleviates Vascular Alterations in Hypertensive Rats"

_medicina, 2020, doi:10.3390/medicina56100509_

Round 1

Reviewer 1 Report

 (General aspect)

The present study suggested that SG extract might be able to restore changes in biochemical and pathological parameters corresponding to vascular dysfunction and remodeling induced by L-NAME administration in consistent with those of Captopril. Therefore, it seemed reasonable to speculate that this extract could cause antihypertensive effect probably through the same mechanism to Captopril.

The experiments were designed without any serious flaw, and carried out without a hitch. Obtained results and given discussion seemed reasonable and understandable. However, the manuscript seemed poor, and there were so many grammatical errors, inadequate use of words, odd or irrelevant expression, unnatural structures of the sentences, inconsistent use of singular or plural form, and so on. Therefore, it would be absolutely necessary to revise radically the manuscript with the help of a native speaker of English.

Considering the length and contents of manuscript, the number of references seemed too many. It would be better to reconsider very carefully and tried to make the explanation simple and clear, thus reducing the references.

(Minor points)

Several examples are given below, and would be anticipated to use as a reference for revising the manuscript.

(Page 1)

  1. Lines 2-3: The word “mitigate” was uncommonly used, and therefore it would be better to use the word “alleviate (or recover, restore)” instead of “mitigate” in the title. In fact, the authors suggested that SG extract caused antihypertensive effect through the alleviation of vascular dysfunction and remodeling. In addition, a singular form “extract” was used here, but a plural form “extracts” was used in many other parts. It would be necessary to consider cautiously which might be relevant.
  2. Lines 17-20: The sentences seemed awkward and unpolished. It would be better to say “the effect of SG on hypertension still remain unknown. Then, the effect of SG aqueous effect on blood pressure and vascular changes were examined n L-NAME-induced hypertensive rats (LHR), and its potential active constituents were also explored”.
  3. Lines 20-24: A detailed explanation of the methods was inappropriate to present in the Abstract section, and it should be described in the Method section.
  4. Line 22: The word “clarified” seemed odd, and it would be better to say “investigated” or “analyzed”.
  5. Lines 24-25: The sentence seemed too simple and insufficient, and it would be better to explain more in details. For example “The analysis of aqueous SG extract was carried out using a HPLC-Mass spectroscopy, and phenolic compounds could be identified as predominant components, which might be responsible to its antihypertensive effect observed in LHR model.
  6. Lines 27-29: The sentence seemed redundant and too long, and it should be simple, clear and easy to understand.
  7. Line 30: The phrase “due to its ability” seemed weird.
  8. Lines 31-32: The effects of Captopril should be explained in comparison with those of SG.
  9. Line 33: The phrase “relevant to” should be “being relevant to”, “as a result of “, or rather simply “through”.

(Page 2)

  1. Lines 50-55: This part would be moved and attached to the end of line 49. From the phrase “Previous studies” on line 55 would be better to be a new paragraph.
  2. Line 56: The word “plays” was wrong, and it should be the past tense.
  3. Lines57-58: The structure of this sentence seemed awkward, and it would be better to say “In L-NAME hypertensive rats, oxidative stress has been reported to increase lipid and protein peroxidation, plasma malondialdehyde (MDA) levels and protein carbonyl contents”.
  4. Line 61: The phrase “Therefore, an agent or plant extract might raise” would be “Therefore, various substances with antioxidant properties could be considered to cause the enhancement of NO bioavailability and the suppression of hypertension.
  5. Line 64: The phrase “South-East Asian” might be a mistake. It should be “South-East Asia”. The phrase “the nutracertical merit” instead of “the nutracertical value” would be better.
  6. Lines 71-72: The sentence seemed quite strange and uneasy, and it would be “SG extract was speculated to have potential anti-cancer activity, due to its cytotoxic effect on cancer cells in vitro.
  7. Lines 72-75: This part seemed awkward and redundant, and also seemed not clear enough to understand.
  8. Line 81: The word “followed” instead of “obeyed” would be better.

(Page 3)

  1. Line 90: The phrase “were administrated orally daily for” seemed awkward, and it would be better to say “were orally administrated every day for”.
  2. Lines 100-101: The structure of this sentence seemed strange, and would be better to write “systolic blood pressure (SBP) and heart rate (HR) were determined in all rats using tail-cuff methods (IITC/Life Science Instrument model 229 and model 179 amplifier (Woodland Hills, CA, USA) throughout 5 weeks of the experiment period.
  3. Line 109: The preposition “in” would probably be “with”.
  4. Line 112: The subheading 2.5 should be “Measurement of oxidative stress markers and plasma nitrate/nitrite”.
  5. Line 113: The phrase “Carotid artery generated superoxide (O2•−) was” seemed odd, and it would be “Superoxide (O2•−) generated in carotid artery was”.
  6. Line 114: The phrase “that followed” should be “following”.
  7. Lines 114-116: The sentence “An enzymatic conversion method as described by Verdon et al 1995 [30] with some modification [29] was used to evaluate plasma NOx levels” has strange structure, and it would be better “Plasma NOx levels were determined using an enzymatic conversion method reported by Verdon et al 1995 [30] with some modification [29].
  8. Lines 116-120: This part was quite awkward, and it would be as follows: The plasma was deproteinized (with perchloric acid, TCA or what?) and the collected supernatant (sample volume?) was mixed with NADPH, G-6-P, G-6-PD and nitrate reductase, and then incubated at 30oC for 30 minutes. The reaction mixture was mixed with a Griess solution (volume?), and kept it at room temperature for 15 min. Then, the absorbance at 540 nm was measured using a microplate reader (Tecan GmbH., Groding Australia).
  9. Line 120: This sentence would be “the serial dilution of NaNO2 stock solution was carried out to prepare standard solutions reported previously [21].
  10. Line 124: The word “being” was not necessary, and should be deleted, and the preposition “with” would be replaced with “and”.

(Page 4)

  1. Line 132: The phrase “reaction sample” was odd, and it should be “reaction mixture”.
  2. Line 135: The word “absorbance” was wrong, and it should be “fluorescence”.
  3. Line 140: The phrase “experimental group” was odd, and it would be “tissues”.
  4. Lines 140-141: The sentence seemed ambiguous and hard to understand, and it would be better to say “The expression of β-actin was also determined as an internal standard”.
  5. Line 143: The phrase “the mean ± the standard error of the mean (SEM) “ was quite redundant, or rather quite strange, and it would be good enough just to say “the mean ± standard error (SEM) “.
  6. Lines 143-144: The phrase “were analyzed using” required a subject, and therefore it would be good to say “the statistical analysis was carried out using GraphPad prism 8.3 software”.
  7. Line 146: The sentence “P < 0.05 was indicated as statistical significance” seemed weird, and it would be “P < 0.05 was considered to be statistically significant.
  8. Line 149: The phrase “Biochemical components of crude extracts” should be “Components of aqueous SG extract”.
  9. Line 151 and 153: The word “features” seemed uncommon way to use, and it would be possible to be replaced with “signals”.
  10. Line 153: The preposition “by” instead of “through” would be better.

(Page 5)

  1. Line 161: The title of table 1 seemed quite strange, might be wrong, and it would be “Major components of aqueous SG extract analyzed using RP-UHPLC-QTOF-MS”.

(Page 6)

  1. Lines 167-168: The phrase “caused high systolic blood pressure” seemed quite odd, and it would be better to say “caused the elevation of systolic blood pressure”.
  2. Line 169: The phrase “Supplementation with SG extract” seemed quite odd, and it would be “Administration of SG extract”
  3. Line 171: The phrase “when compared with untreated rats” was unnecessary, because SG was just shown to reduce blood pressure, but the effect of SG was not compared to any other, statistically or non-statistically.
  4. Line 172: The phrase “compared to control rats” was unnecessary and should be deleted. Also, the phrase “at week 5, 549.94 ± 12.38 vs 425.19 ± 4.51 beats/min, p<0.05)” was unclear, and it would be “from 425.19 ± 4.51 to 549.94 ± 12.38 beats/min (p<0.05) at week 5.
  5. Line 175: The value “at 474.75 ± 19.56 beats/min, p<0.05” would be better to put in parenthesis without “at”, just like (474.75 ± 19.56 beats/min, p<0.05).
  6. Lines 181-182: The sentence “The vasorelaxation response to ACh (0.01 - 3 mM) was significantly blunted in aortic rings” was odd, and it would be better to say “The vasorelaxation response of aortic rings to ACh (0.01 - 3 mM) was significantly attenuated in LHRs”.
  7. Line 185: The phrase “not different” would be better to replace with “almost similar”, or use other expression.

(Page 7)

  1. Line 189: The word “Effects” should be a singular form.
  2. Line 197: The word “attenuation” seemed strange term here, and it would be replaced with “reduction”.
  3. Line 200: The phrase “Effects of SG extracts” might be “Effect of SG extract” (singular).

(Page 8)

  1. Line 205: The word “alleviated” would be replaced with other word, for example “suppressed”
  2. Lines 206-208: The structure of this sentence seemed unnatural, and would be revised. For example. Significant increases in the indices of vascular remodeling, such as cross-sectional area (CSA), wall thickness, lumen diameter, wall thickness/lumen ratio and VSMC were observed in thoracic aortas obtained from LHRs.
  3. Line 209: The word “improved” could be replaced with “alleviated”.
  4. Line 211: The phrase “Aortic hypertrophic morphology” was quite unnatural and weird, and there might be no such expression. Therefore, the phrase should be rewritten. For example, “Morphology of aortic hypertrophy” or “Morphological change in aortic hypertrophy”.
  5. Line 216: Which was correct, singular (level) or plural (levels).
  6. Line 218: The word “supplementation” seemed unnecessary, and should be deleted. In addition, the word “increases” should be singular (increase).

(Page 9)

  1. Line 233: The word “production” was not wrong, but better to be replaced with “generation”.
  2. Line 235: The word “attenuated” could be good to be replaced with “restored” or “recovered”
  3. Line 239: The word “Effects” would be singular.
  4. Line 243: The sentence “This study found that the main components of SG extract are phenolic compounds” seemed awkward and poor, and it would be better to say “This study suggested that the major components of SG extract might be phenolic compounds” or “This study could proposed phenolic compounds as major components of aqueous SG extract”.
  5. Lines 243-245: The sentence “SG extract produced a dose dependent manner in reduction of blood pressure in rats with chronic administration of L-NAME to induced hypertension” seemed quite odd, and the structure of this sentence might be wrong. It would probably be good to say “SG extract caused the reduction of blood pressure in a dose-dependent manner in hypertensive rat model produced by chronic L-NAME administration.
  6. Lines 245- line 248 (page 10): The sentence was redundant and unclear, and not so easy to read and understand. It should be revised as follows: First of all, the phrase “supported by increases 246 in CSA, wall thickness, wall/lumen ratio and smooth muscle cell numbers” should be deleted. Then, the phrase “in aortic rings” would be moved at the end of line 247, and followed by “prepared from”.

The revised sentence was like this: Endothelial dysfunction characterized by a decrease in vascular response to Ach and aortic hypertrophy were observed in aortic rings prepared from hypertensive rats.

(Page 10)

  1. Lines 259-260: The sentence “In this study, bioactive substances, flavonoids listed in table 1 are abundant in SG extract” was supposed to say that flavonoids were abundantly contained in SG extract. If so, this sentence should be “In this study, bioactive substances, flavonoids listed in table 1, were shown to be abundant in SG extract” or “this study showed that bioactive substances, flavonoids listed in table 1, were abundantly contained in SG extract”.
  2. Line 260: The phrase “health benefit” was good, but “health maintenance” or simply “human health” would be better.
  3. Lines 262-263: The phrase “that are relevant to its active substance” seemed unclear, rather hard to understand the meaning. Would it be possible to delete?
  4. Line 264: The phrase “high blood pressure and HR” should be “high blood pressure and increased HR”.
  5. Line 266: The word “reductions” should be singular “reduction”.
  6. Line 279: What was “SP”?
  7. Line 281: The phrase “ACE activity and serum Ang II as well as oxidative stress markers” seemed incomplete and was lacking some word. It should be “the elevation of ACE activity and serum Ang II as well as oxidative stress markers”.
  8. Line 282: The phrase “a reduction of RAS activation” would be better to say “the suppression of RAS activation”. The activation was generally suppressed or inhibited, but not reduced.
  9. Lines 282-283: The phrase “a consequence of ACE inhibitory activity” was quite awkward. In order to emphasize a consequence, “ACE inhibitory activity” should be “ACE inhibition”. On the other hand, to suggest ACE inhibitory activity, “a consequence of” should be “due to the ACE inhibitory activity of SG extract”.
  10. Line 285: The phrase “that suggested” would be just “suggesting”.
  11. Line 286: The phrase “stop its activity” should be “stop the enzyme reaction”.
  12. Line 287: The word “involved” should be followed by “in”.
  13. Lines 297-298: The sentence “This section is mandatory and should contain the main conclusions regarding the research” was unnecessary and should be deleted.

Author Response

Dear Professor,

We would like to thank reviewer for valuable comments and suggestion. Your comments and suggestions make the manuscript more complete. Responses to your comments have been carefully done point-by-point as follows;

Comments and Suggestions for Authors

 (General aspect)

The present study suggested that SG extract might be able to restore changes in biochemical and pathological parameters corresponding to vascular dysfunction and remodeling induced by L-NAME administration in consistent with those of Captopril. Therefore, it seemed reasonable to speculate that this extract could cause antihypertensive effect probably through the same mechanism to Captopril.

Response: We agree with reviewer suggestion. We added this sentence “Captopril produced antihypertensive effect via alleviating vascular dysfunction and remodeling in LHRs. Additionally, captopril also reduced oxidative stress and RAS activation in LHRs. Therefore, it is possible that SG extract had antihypertensive effect probably through the same mechanism to captopril” at the end of the last paragraph of discussion.

The experiments were designed without any serious flaw, and carried out without a hitch. Obtained results and given discussion seemed reasonable and understandable. However, the manuscript seemed poor, and there were so many grammatical errors, inadequate use of words, odd or irrelevant expression, unnatural structures of the sentences, inconsistent use of singular or plural form, and so on. Therefore, it would be absolutely necessary to revise radically the manuscript with the help of a native speaker of English.

Response: The English has been improved by Professor James A. Will.

Considering the length and contents of manuscript, the number of references seemed too many. It would be better to reconsider very carefully and tried to make the explanation simple and clear, thus reducing the references.

Response: The number of references has been reduced as reviewer suggested.

(Minor points)

Several examples are given below, and would be anticipated to use as a reference for revising the manuscript.

Response to Reviewer 1 Comments

Point 1: Lines 2-3: The word “mitigate” was uncommonly used, and therefore it would be better to use the word “alleviate (or recover, restore)” instead of “mitigate” in the title. In fact, the authors suggested that SG extract caused antihypertensive effect through the alleviation of vascular dysfunction and remodeling. In addition, a singular form “extract” was used here, but a plural form “extracts” was used in many other parts. It would be necessary to consider cautiously which might be relevant.

Response 1: The word mitigate has been replaced with alleviate. “extract” has been used in all part of the text.

Point 2: Lines 17-20: The sentences seemed awkward and unpolished. It would be better to say “the effect of SG on hypertension still remain unknown. Then, the effect of SG aqueous effect on blood pressure and vascular changes were examined n L-NAME-induced hypertensive rats (LHR), and its potential active constituents were also explored”.

Response 2: The sentences in line 17-20 has been changed to “the effect of SG on hypertension still remain unknown. Then, the effect of SG aqueous extract on blood pressure and vascular changes were examined n L-NAME-induced hypertensive rats (LHR), and its potential active constituents were also explored” as reviewer suggested.

Point 3: Lines 20-24: A detailed explanation of the methods was inappropriate to present in the Abstract section, and it should be described in the Method section.

Response 3: Line 20-24 has been modified as “Male Sprague Dawley rats were allocated to control, L-NAME (40 mg/kg/day), L-NAME + SG (100, 300, 500 mg/kg/day) or captopril (5 mg/kg/day) groups. The components of SG extract were analyzed”.

Point 4: Line 22: The word “clarified” seemed odd, and it would be better to say “investigated” or “analyzed”.

Response 4: The word “clarified” has been changed to analyzed.

Point 5: Lines 24-25: The sentence seemed too simple and insufficient, and it would be better to explain more in details. For example “The analysis of aqueous SG extract was carried out using a HPLC-Mass spectroscopy, and phenolic compounds could be identified as predominant components, which might be responsible to its antihypertensive effect observed in LHR model.

Response 5: The sentences in line 24-25 has been changed to “The analysis of aqueous SG extract was carried out using a HPLC-Mass spectroscopy, and phenolic compounds could be identified as predominant components, which might be responsible to its antihypertensive effect observed in LHR model”.

Point 6: Lines 27-29: The sentence seemed redundant and too long, and it should be simple, clear and easy to understand.

Response 6: The sentences in line 27-29 has been modified to “Enhancements of eNOS expression and plasma nitric oxide metabolite levels, and attenuation of angiotensin converting enzyme (ACE) activity and plasma angiotensin II levels were observed in the LHR group treated with SG”.

Point 7: Line 30: The phrase “due to its ability” seemed weird.

Response 7: The phrase “due to its ability” has been changed to by reducing.

Point 8: Lines 31-32: The effects of Captopril should be explained in comparison with those of SG.

Response 8: The sentences in lines 31-32 has been modified to “Captopril suppressed high blood pressure and alleviated vascular changes and ACE activity in LHRs similar to those of SG extracts (p<0.05).

Point 9: Line 33: The phrase “relevant to” should be “being relevant to”, “as a result of “, or rather simply “through”.

Response 9: The phrase “relevant to” has been replaced with being relevant to.

Point 10: Lines 50-55: This part would be moved and attached to the end of line 49. From the phrase “Previous studies” on line 55 would be better to be a new paragraph.

Response 10: Point 10 has been changed as reviewer suggested.

Point 11: Line 56: The word “plays” was wrong, and it should be the past tense.

Response 11: The word “plays” has been changed to played.

Point 12: Lines57-58: The structure of this sentence seemed awkward, and it would be better to say “In L-NAME hypertensive rats, oxidative stress has been reported to increase lipid and protein peroxidation, plasma malondialdehyde (MDA) levels and protein carbonyl contents”.

Response 12: Point 12 has been changed as reviewer suggested.

Point 13: Line 61: The phrase “Therefore, an agent or plant extract might raise” would be “Therefore, various substances with antioxidant properties could be considered to cause the enhancement of NO bioavailability and the suppression of hypertension.

Response 13: Point 13 has been changed as reviewer suggested.

Point 14: Line 64: The phrase “South-East Asian” might be a mistake. It should be “South-East Asia”. The phrase “the nutracertical merit” instead of “the nutracertical value” would be better.

Response 14: Point 14 has been changed as reviewer suggested.

Point 15: Lines 71-72: The sentence seemed quite strange and uneasy, and it would be “SG extract was speculated to have potential anti-cancer activity, due to its cytotoxic effect on cancer cells in vitro.

Response 15: Lines 71-72 has been changed to “SG extract was speculated to have potential anti-cancer activity, due to its cytotoxic effect on cancer cells in vitro” as reviewer suggested.

Point 16: Lines 72-75: This part seemed awkward and redundant, and also seemed not clear enough to understand.

Response 16: Lines 72-75 has been modified to “A recent study showed that  phenolic compounds in SG methanolic extract might mediate cytotoxicity activity in MCF-7 breast adenocarcinoma and MDA-MB-231 breast cancer cell lines [26]”.

Point 17: Line 81: The word “followed” instead of “obeyed” would be better.

Response 17: The word “followed” has been replaced with “obeyed”.

Point 18: Line 90: The phrase “were administrated orally daily for” seemed awkward, and it would be better to say “were orally administrated every day for”.

Response 18: The phrase “were administrated orally daily for” has been changed to “were orally administrated every day for” as reviewer suggested.

Point 19: Lines 100-101: The structure of this sentence seemed strange, and would be better to write “systolic blood pressure (SBP) and heart rate (HR) were determined in all rats using tail-cuff methods (IITC/Life Science Instrument model 229 and model 179 amplifier (Woodland Hills, CA, USA) throughout 5 weeks of the experiment period.

Response 19: The sentences in lines 100-101 has been modified to “systolic blood pressure (SBP) and heart rate (HR) were determined in all rats using tail-cuff methods (IITC/Life Science Instrument model 229 and model 179 amplifier (Woodland Hills, CA, USA) throughout 5 weeks of the experiment period” as reviewer suggested.

Point 20: Line 109: The preposition “in” would probably be “with”.

Response 20: The preposition “in” has been replaced with “with”.

Point 21: Line 112: The subheading 2.5 should be “Measurement of oxidative stress markers and plasma nitrate/nitrite”.

Response 21: The subheading 2.5 has been changed to “Measurement of oxidative stress markers and plasma nitrate/nitrite”.

Point 22: Line 113: The phrase “Carotid artery generated superoxide (O2•−) was” seemed odd, and it would be “Superoxide (O2•−) generated in carotid artery was”.

Response 22:  The phrase “Carotid artery generated superoxide (O2•−) has been changed to “Superoxide (O2•−) generated in carotid artery was”.

Point 23: Line 114: The phrase “that followed” should be “following”.

Response 23:  The phrase “that followed” has been changed to “following”.

Point 24: Lines 114-116: The sentence “An enzymatic conversion method as described by Verdon et al 1995 [30] with some modification [29] was used to evaluate plasma NOx levels” has strange structure, and it would be better “Plasma NOx levels were determined using an enzymatic conversion method reported by Verdon et al 1995 [30] with some modification [29].

Response 24:  The sentences in lines 114-116 has been changed to “Plasma NOx levels were determined using an enzymatic conversion method reported by Verdon et al 1995 [30] with some modification [29].

Point 25: Lines 116-120: This part was quite awkward, and it would be as follows: The plasma was deproteinized (with perchloric acid, TCA or what?) and the collected supernatant (sample volume?) was mixed with NADPH, G-6-P, G-6-PD and nitrate reductase, and then incubated at 30oC for 30 minutes. The reaction mixture was mixed with a Griess solution (volume?), and kept it at room temperature for 15 min. Then, the absorbance at 540 nm was measured using a microplate reader (Tecan GmbH., Groding Australia).

Response 25:  The sentences in lines 116-120 has been modified to “the plasma protein was filtrated using nanosep (Pall Life sciences, Portsmouth, UK) and the filtrate (80 µl) was collected and mixed with NADPH, G-6-P, G-6-PD and nitrate reductase, and then incubated at 30°C for 30 minutes. The reaction mixture was mixed with a Griess solution (100 µl), and kept it at room temperature for 15 min. Then, the absorbance at 540 nm was measured using a microplate reader (Tecan GmbH., Groding Australia).

Point 26: Line 120: This sentence would be “the serial dilution of NaNO2 stock solution was carried out to prepare standard solutions reported previously [21].

Response 26: The sentences in line 120 has been changed to “the serial dilution of NaNO2 stock solution was carried out to prepare standard solutions reported previously [21].

Point 27: Line 124: The word “being” was not necessary, and should be deleted, and the preposition “with” would be replaced with “and”.

Response 27: Line 124 has been changed as reviewer suggested.

Point 28: Line 132: The phrase “reaction sample” was odd, and it should be “reaction mixture”.

Response 28: Line 132 has been changed as reviewer suggested.

Point 29: Line 135: The word “absorbance” was wrong, and it should be “fluorescence”.

Response 29: Line 135 has been changed as reviewer suggested.

Point 30: Line 140: The phrase “experimental group” was odd, and it would be “tissues”.

Response 30: Line 140 has been changed as reviewer suggested.

Point 31: Lines 140-141: The sentence seemed ambiguous and hard to understand, and it would be better to say “The expression of β-actin was also determined as an internal standard”.

Response 31: Line 140-141 has been changed as reviewer suggested.

Point 32: Line 143: The phrase “the mean ± the standard error of the mean (SEM) “ was quite redundant, or rather quite strange, and it would be good enough just to say “the mean ± standard error (SEM)”.

Response 32: Line 143 has been changed as reviewer suggested.

Point 33: Lines 143-144: The phrase “were analyzed using” required a subject, and therefore it would be good to say “the statistical analysis was carried out using GraphPad prism 8.3 software”.

Response 33: Line 143-144 has been changed as reviewer suggested.

Point 34: Line 146: The sentence “P < 0.05 was indicated as statistical significance” seemed weird, and it would be “P < 0.05 was considered to be statistically significant.

Response 34: Line 146 has been changed as reviewer suggested.

Point 35: Line 149: The phrase “Biochemical components of crude extracts” should be “Components of aqueous SG extract”.

Response 35: Line 149 has been changed as reviewer suggested.

Point 36: Line 151 and 153: The word “features” seemed uncommon way to use, and it would be possible to be replaced with “signals”.

Response 36: The word “features” has been replaced with “signals”.

Point 37: Line 153: The preposition “by” instead of “through” would be better.

Response 37: The preposition “through” has been replaced by “by”.

Point 38: Line 161: The title of table 1 seemed quite strange, might be wrong, and it would be “Major components of aqueous SG extract analyzed using RP-UHPLC-QTOF-MS”.

Response 38: The title of table 1 has been changed as reviewer suggested.

Point 39: Lines 167-168: The phrase “caused high systolic blood pressure” seemed quite odd, and it would be better to say “caused the elevation of systolic blood pressure”.

Response 39: Line 167-168 has been changed as reviewer suggested.

Point 40: Line 169: The phrase “Supplementation with SG extract” seemed quite odd, and it would be “Administration of SG extract”

Response 40: Line 169 has been changed as reviewer suggested.

Point 41: Line 171: The phrase “when compared with untreated rats” was unnecessary, because SG was just shown to reduce blood pressure, but the effect of SG was not compared to any other, statistically or non-statistically.

Response 41: Line 171 has been changed as reviewer suggested.

Point 42: Line 172: The phrase “compared to control rats” was unnecessary and should be deleted. Also, the phrase “at week 5, 549.94 ± 12.38 vs 425.19 ± 4.51 beats/min, p<0.05)” was unclear, and it would be “from 425.19 ± 4.51 to 549.94 ± 12.38 beats/min (p<0.05) at week 5.

Response 42: Line 172 has been changed as reviewer suggested.

Point 43: Line 175: The value “at 474.75 ± 19.56 beats/min, p<0.05” would be better to put in parenthesis without “at”, just like (474.75 ± 19.56 beats/min, p<0.05).

Response 43: Line 175 has been changed as reviewer suggested.

Point 44: Lines 181-182: The sentence “The vasorelaxation response to ACh (0.01 - 3 mM) was significantly blunted in aortic rings” was odd, and it would be better to say “The vasorelaxation response of aortic rings to ACh (0.01 - 3 mM) was significantly attenuated in LHRs”.

Response 44: Line 181-182 has been changed as reviewer suggested.

Point 45: Line 185: The phrase “not different” would be better to replace with “almost similar”, or use other expression.

Response 45: Line 185: The phrase “not different” has been replaced with “almost similar”.

Point 46: Line 189: The word “Effects” should be a singular form.

Response 46: Line 189 has been changed as reviewer suggested.

Point 47: Line 197: The word “attenuation” seemed strange term here, and it would be replaced with “reduction”.

Response 47: Line 197 has been changed as reviewer suggested.

Point 48: Line 200: The phrase “Effects of SG extracts” might be “Effect of SG extract” (singular).

Response 48: Line 200 has been changed as reviewer suggested.

Point 49: Line 205: The word “alleviated” would be replaced with other word, for example “suppressed”

Response 49: Line 205 has been changed as reviewer suggested.

Point 50: Lines 206-208: The structure of this sentence seemed unnatural, and would be revised. For example. Significant increases in the indices of vascular remodeling, such as cross-sectional area (CSA), wall thickness, lumen diameter, wall thickness/lumen ratio and VSMC were observed in thoracic aortas obtained from LHRs.

Response 50: Line 206-208 has been revised as reviewer suggested.

Point 51: Line 209: The word “improved” could be replaced with “alleviated”.

Response 51: The word “improved” has been replaced with “alleviated”.

Point 52: Line 211: The phrase “Aortic hypertrophic morphology” was quite unnatural and weird, and there might be no such expression. Therefore, the phrase should be rewritten. For example, “Morphology of aortic hypertrophy” or “Morphological change in aortic hypertrophy”.

Response 52: Line 211 has been changed as reviewer suggested.

Point 53: Line 216: Which was correct, singular (level) or plural (levels).

Response 53: Line 216: It should be singular and it has been corrected in the text.

Point 54: Line 218: The word “supplementation” seemed unnecessary, and should be deleted. In addition, the word “increases” should be singular (increase).

Response 54: Line 218 has been changed as reviewer suggested.

Point 55: Line 233: The word “production” was not wrong, but better to be replaced with “generation”.

Response 55: Line 218 has been changed as reviewer suggested.

Point 56: Line 235: The word “attenuated” could be good to be replaced with “restored” or “recovered”

Response 56: Line 235 The word “attenuated” has been replaced with restored.

Point 57: Line 239: The word “Effects” would be singular.

Response 57: Line 239 has been changed as reviewer suggested.

Point 58: Line 243: The sentence “This study found that the main components of SG extract are phenolic compounds” seemed awkward and poor, and it would be better to say “This study suggested that the major components of SG extract might be phenolic compounds” or “This study could proposed phenolic compounds as major components of aqueous SG extract”.

Response 58: Line 243 has been changed as reviewer suggested.

Point 59: Lines 243-245: The sentence “SG extract produced a dose dependent manner in reduction of blood pressure in rats with chronic administration of L-NAME to induced hypertension” seemed quite odd, and the structure of this sentence might be wrong. It would probably be good to say “SG extract caused the reduction of blood pressure in a dose-dependent manner in hypertensive rat model produced by chronic L-NAME administration.

Response 59: Line 243-245 has been changed as reviewer suggested.

Point 60: Lines 245- line 248 (page 10): The sentence was redundant and unclear, and not so easy to read and understand. It should be revised as follows: First of all, the phrase “supported by increases 246 in CSA, wall thickness, wall/lumen ratio and smooth muscle cell numbers” should be deleted. Then, the phrase “in aortic rings” would be moved at the end of line 247, and followed by “prepared from”.

The revised sentence was like this: Endothelial dysfunction characterized by a decrease in vascular response to Ach and aortic hypertrophy were observed in aortic rings prepared from hypertensive rats.

Response 60: Line 245-248 has been revised as reviewer suggested.

Point 61: Lines 259-260: The sentence “In this study, bioactive substances, flavonoids listed in table 1 are abundant in SG extract” was supposed to say that flavonoids were abundantly contained in SG extract. If so, this sentence should be “In this study, bioactive substances, flavonoids listed in table 1, were shown to be abundant in SG extract” or “this study showed that bioactive substances, flavonoids listed in table 1, were abundantly contained in SG extract”.

Response 61: Line 259-260 has been revised as reviewer suggested.

Point 62: Line 260: The phrase “health benefit” was good, but “health maintenance” or simply “human health” would be better.

Response 62: Line 260 has been changed as reviewer suggested.

Point 63: Lines 262-263: The phrase “that are relevant to its active substance” seemed unclear, rather hard to understand the meaning. Would it be possible to delete?

Response 63: Line 262-263: The phrase “that are relevant to its active substance” has been removed as reviewer suggested.

Point 64: Line 264: The phrase “high blood pressure and HR” should be “high blood pressure and increased HR”.

Response 63: Line 264 has been changed as reviewer suggested.

Point 65: Line 266: The word “reductions” should be singular “reduction”.

Response 65: Line 266 has been changed as reviewer suggested.

Point 66: Line 279: What was “SP”?

Response 66: SP has been changed to systolic blood pressure.

Point 67: Line 281: The phrase “ACE activity and serum Ang II as well as oxidative stress markers” seemed incomplete and was lacking some word. It should be “the elevation of ACE activity and serum Ang II as well as oxidative stress markers”.

Response 67: Line 281 has been changed as reviewer suggested.

Point 68: Line 282: The phrase “a reduction of RAS activation” would be better to say “the suppression of RAS activation”. The activation was generally suppressed or inhibited, but not reduced.

Response 68: Line 282 has been changed as reviewer suggested.

Point 69: Lines 282-283: The phrase “a consequence of ACE inhibitory activity” was quite awkward. In order to emphasize a consequence, “ACE inhibitory activity” should be “ACE inhibition”. On the other hand, to suggest ACE inhibitory activity, “a consequence of” should be “due to the ACE inhibitory activity of SG extract”.

Response 69: Line 282-283 has been changed as reviewer suggested.

Point 70: Line 285: The phrase “that suggested” would be just “suggesting”.

Response 70: Line 285 has been changed as reviewer suggested.

Point 71: Line 286: The phrase “stop its activity” should be “stop the enzyme reaction”.

Response 71: Line 286 has been changed as reviewer suggested.

Point 72: Line 287: The word “involved” should be followed by “in”.

Response 72: Line 287 has been changed as reviewer suggested.

Point 73: Lines 297-298: The sentence “This section is mandatory and should contain the main conclusions regarding the research” was unnecessary and should be deleted.

Response 73: The sentence “This section is mandatory and should contain the main conclusions regarding the research” has been removed.

Thank you very much for carefully reading our manuscript, the comment and suggestion are valuable that make our manuscript more complete.

Sincerely yours,

Putcharawipa Maneesai

Reviewer 2 Report

In this manuscript, authors investigated the role of Syzygium gratum extract, in mitigating the  vascular alterations in L-name, a NOS inhibitor induced hypertensive rat model. In this work authors used the L-name to induce the hypertension and ingested the S. gratum extract to test if this can mitigate the vascular alterations. Using tail-cuff method and organ chamber system, they tested the experimental results.  

Provided data is enough to validate the results. 

However, some minor concerns need to be answered.

  1. Figure 2b, looks like compressed compared to the Figure 2a.
  2. Labelling/text on Figures (all of the diagrams are not clear enough. Magnify the text and replot the graphs.
  3. Some texts need English improvement :

For ex.  “Firstly, the plasma was deproteinized and the supernatant collected’. –Doesn’t sounds perfect, Need improvement.

  1. Many typos and Grammar need to be improved throughout the text.

Author Response

Dear Professor,

We would like to thank reviewer for valuable comments and suggestion. Responses to your comments have been carefully done point-by-point as follows;

Response to Reviewer 2

Comments and Suggestions for Authors

In this manuscript, authors investigated the role of Syzygium gratum extract, in mitigating the  vascular alterations in L-name, a NOS inhibitor induced hypertensive rat model. In this work authors used the L-name to induce the hypertension and ingested the S. gratum extract to test if this can mitigate the vascular alterations. Using tail-cuff method and organ chamber system, they tested the experimental results. 

Provided data is enough to validate the results.

Response: Thank you very much for your comments

However, some minor concerns need to be answered.

Point 1: Figure 2b, looks like compressed compared to the Figure 2a.

Response 1: The figure 2b has been uncompressed and improved.

Point 2: Labelling/text on Figures (all of the diagrams are not clear enough. Magnify the text and replot the graphs.

Response 2: The quality of the figures has been improved as reviewer suggested.

Point 3: Some texts need English improvement:

For ex.  “Firstly, the plasma was deproteinized and the supernatant collected’. –Doesn’t sounds perfect, Need improvement.

Response 3: The English has been improved by Prof. James A. Will as track change marks.

Point 4: Many typos and Grammar need to be improved throughout the text.

Response 4: The English has been improved by Prof. James A. Will.

Thank you very much for carefully reading our manuscript, the comment and suggestion are valuable that make our manuscript more complete.

Sincerely yours,

Putcharawipa Maneesai

Round 2

Reviewer 1 Report

The manuscript was totally revised and improved, and it now became much clearer and easier to read and understand. As mentioned in the previous comments, the experiments were designed without any serious flaw, and carried out without a hitch, and the obtained results and given discussion therefore seemed reasonable and reliable. Nevertheless, this work is not considered to have a big impact scientifically and clinically on hypertension treatment. However, it would be quite meaningful to accumulate such reliable findings for the progress of antihypertensive drug development. In future research, it would be very much to be desired that the effect of SG extract on blood pressure and vascular function would also be studied using different types of hypertension models.